# Hydrogen Peroxide Stimulates Dihydrotestosterone Release in C2C12 Myotubes: A New Perspective for Exercise-Related Muscle Steroidogenesis?

**DOI:** 10.3390/ijms23126566

**Published:** 2022-06-12

**Authors:** Cristina Antinozzi, Guglielmo Duranti, Roberta Ceci, Marco Lista, Stefania Sabatini, Daniela Caporossi, Luigi Di Luigi, Paolo Sgrò, Ivan Dimauro

**Affiliations:** 1Endocrinology Unit, Department of Movement, Human and Health Sciences, Università degli Studi di Roma “Foro Italico”, Piazza Lauro De Bosis 6, 00135 Roma, Italy; cristina.antinozzi@uniroma4.it (C.A.); m.lista@studenti.uniroma4.it (M.L.); luigi.diluigi@uniroma4.it (L.D.L.); 2Laboratory of Biochemistry of Movement, Department of Movement, Human and Health Sciences, Università degli Studi di Roma “Foro Italico”, Piazza Lauro De Bosis 6, 00135 Roma, Italy; roberta.ceci@uniroma4.it (R.C.); stefania.sabatini@uniroma4.it (S.S.); 3Laboratory of Biology and Human Genetic, Department of Movement, Human and Health Sciences, Università degli Studi di Roma “Foro Italico”, Piazza Lauro De Bosis 6, 00135 Roma, Italy; daniela.caporossi@uniroma4.it (D.C.); ivan.dimauro@uniroma4.it (I.D.)

**Keywords:** skeletal muscles, reactive oxygen species, redox status, testosterone, dihydrotestosterone, tadalafil, phosphodiesterase type 5

## Abstract

Skeletal muscle is a tissue that has recently been recognized for its ability to produce androgens under physiological conditions. The steroidogenesis process is known to be negatively influenced by reactive oxygen species (ROS) in reproductive Leydig and ovary cells, while their effect on muscle steroidogenesis is still an unexplored field. Muscle cells are continuously exposed to ROS, resulting from both their metabolic activity and the surrounding environment. Interestingly, the regulation of signaling pathways, induced by mild ROS levels, plays an important role in muscle fiber adaptation to exercise, in a process that also elicits a significant modulation in the hormonal response. The aim of the present study was to investigate whether ROS could influence steroidogenesis in skeletal muscle cells by evaluating the release of testosterone (T) and dihydrotestosterone (DHT), as well as the evaluation of the relative expression of the key steroidogenic enzymes 5α-reductase, 3β-hydroxysteroid dehydrogenase (HSD), 17β-HSD, and aromatase. C2C12 mouse myotubes were exposed to a non-cytotoxic concentration of hydrogen peroxide (H_2_O_2_), a condition intended to reproduce, in vitro, one of the main stimuli linked to the process of homeostasis and adaptation induced by exercise in skeletal muscle. Moreover, the influence of tadalafil (TAD), a phosphodiesterase 5 inhibitor (PDE5i) originally used to treat erectile dysfunction but often misused among athletes as a “performance-enhancing” drug, was evaluated in a single treatment or in combination with H_2_O_2_. Our data showed that a mild hydrogen peroxide exposure induced the release of DHT, but not T, and modulated the expression of the enzymes involved in steroidogenesis, while TAD treatment significantly reduced the H_2_O_2_-induced DHT release. This study adds a new piece of information about the adaptive skeletal muscle cell response to an oxidative environment, revealing that hydrogen peroxide plays an important role in activating muscle steroidogenesis.

## 1. Introduction

The most important male sex hormone, testosterone (T), is mainly produced in the testis by Leydig cells, starting from the precursor, dehydroepiandrosterone (DHEA), generally in response to the luteinizing hormone (LH), and to the activation of the hypothalamic–pituitary–gonadal axis [1,2]. Dihydrotestosterone (DHT), a metabolite of T, has more androgenic effects than T and is produced by peripheral tissues, mainly the prostate. Interestingly, in addition to the testis, the expression of hydroxysteroid-dehydrogenase enzymes has also been found in other tissues [3]. Particularly, the production of testosterone and DHT, independently from the release of LH, has been observed in both mouse and human skeletal muscle [4].

Endogenous androgens can influence cell metabolism and protein synthesis, depending on the time of action and physiological conditions. Moreover, their pattern of production and release can be modified in response to acute or chronic muscle contractions [5,6,7,8,9], through a positive regulatory effect on the expression of the key steroidogenic enzymes. Particularly in peripheral tissues, testosterone is synthesized through the metabolism of DHEA via the action of 3β-hydroxysteroid dehydrogenase (3β-HSD) and 17β-hydroxysteroid dehydrogenase (17β-HSD), and in turn, it is converted into DHT by the irreversible action of 5α-reductase (5α-R), or in estradiol by aromatase (Cyp-19) activity (Figure 1).

An important occurrence observed during muscle contraction is the rise in superoxide anions formation, a process mainly induced by the acceleration in oxidative metabolism, necessary for supporting the major energy demand. These molecules are rapidly converted to hydrogen peroxide (H_2_O_2_), whose relevance, at a low level, in activating redox signaling pathways and transcription factors involved in the adaptive response to exercise, is recognized by a growing number of studies [10,11,12,13,14,15]. The formation of small amounts of ROS acting as redox signaling messengers, needed for the normal physiological functions, is known as “eustress”. As a consequence of this phenomenon, the induction of numerous ROS response proteins is observed. Among them, the small heat shock protein (sHSP), αB-crystallin (CRYAB), is currently considered a sensor of oxidative stress in mammalian cells, with a key role in the prevention of apoptosis [16].

On the contrary, overwhelming concentrations of ROS determine “oxidative distress”. In particular, prolonged exposure to cytotoxic levels of ROS can modulate specific markers of apoptosis (e.g., Bcl-2, Bax, and cleaved caspase-3), leading to programmed cell death [17]. Oxidative stress is a common factor in several age-related physiopathological conditions, and it is widely recognized as a detrimental factor for physical performance [13].

To overcome the negative effects of the excessive ROS amount that is produced following intense exercise, sports practitioners commonly use antioxidant supplementation in sports fields, along with “performer-enhancing” drugs, in the ever-growing attempt to improve their physical efficiency.

Among the non-prohibited substances utilized to optimize sports performance, tadalafil is often misused. This drug belongs to a group of phosphodiesterase type 5 enzyme inhibitors (PDE5i) originally prescribed for the treatment of erectile dysfunction (ED) [18,19].

In addition to its numerous physiological effects, it has been observed that tadalafil also targets muscle tissue, regulating its endocrine–metabolic functions, probably through the modifications of steroid hormone release [20,21]. However, the relationship between TAD, steroid hormones, pro-oxidant environment, and the relative molecular mechanisms are still not completely clarified.

The aim of the present study was to evaluate the effect of H_2_O_2_ on steroidogenesis in skeletal muscle. C2C12 myotubes were treated with a non-cytotoxic concentration of H_2_O_2_ to mimic a eustress condition and analyzed for DHT and T release; the gene expression levels of the steroidogenic enzymes 5α-R, 3β-HSD, 17β-HSD, and aromatase, as well as the protein expression/activation of stress response proteins such as alpha B-crystallin (CRYAB), and apoptotic markers, such as Bcl-2, Bax, and caspase-3. Moreover, we verified whether tadalafil could interfere with the H_2_O_2_-induced cellular response.

## 2. Results

### 2.1. Analysis of Stress Response Protein and Apoptotic Markers

C2C12 cells, treated for 24 h with H_2_O_2_ 500 µM, were analyzed for morphological changes. Figure 2A shows that, in comparison to control cells (left panel, a), the morphological condition of treated cells (right panel, b) was apparently similar, and no cell death was observed. Then, we assayed the effect of hydrogen peroxide treatment on the expression/phosphorylation level of CRYAB in C2C12 myotubes (Figure 2B), as well as on the expression of specific apoptotic markers, such as Bcl-2, Bax, and caspase-3 (Figure 2B–D).

As expected, we observed a significant increase in CRYAB activation (*p*-CRYAB) following exposure to hydrogen peroxide (1 h and 24 h, respectively, 1.8 ± 0.0 and 3.2 ± 0.1-fold change vs. control, *p* < 0.05). Pre-treatment with tadalafil (1 µM) significantly reduced CRYAB activation after 1 h (H_2_O_2_ + TAD: 0.7 ± 0.1-fold change vs. H_2_O_2_, *p* < 0.05). No significant effects on *p*-CRYAB were detected in the presence of tadalafil alone. No differences were observed in control cells at both experimental points (1 h and 24 h; *p* > 0.05) (Figure 2C,D).

No effects on the total content of CRYAB were observed at any experimental condition (*p* > 0.05).

Similarly, no changes were observed for all apoptotic markers analyzed at each experimental condition (*p* > 0.05) (Figure 2B).

### 2.2. Dihydrotestosterone (DHT) and Testosterone (T) Release

We evaluated the effect of hydrogen peroxide treatment on DHT and T release in skeletal muscle. The treatment with H_2_O_2_ (500 µM, 24 h) induced a significant release of DHT compared with control cells (15.6 ± 2.5 vs. 5.1 ± 2.8 pg/mL, *p* < 0.01), whereas no significant effects were observed following tadalafil exposure in comparison to control, although an increasing trend was demonstrated (1 µM, 24 h) (Figure 3A, *p* > 0.05).

The pre-treatment with tadalafil blunted the effect of H_2_O_2_ (9.2 ± 2.8, *p* < 0.05 pg/mL, Figure 3A) by reducing the extent of the phenomenon. However, the DHT release was still significantly higher compared with control cells.

No significant effects were observed on testosterone secretion following each treatment (*p* > 0.05, Figure 3B).

### 2.3. mRNA Expression of Steroidogenic Enzymes

We examined the effect of hydrogen peroxide treatment on the expression of the steroidogenic enzymes mainly expressed in skeletal muscle [22]. H_2_O_2_ (500 µM H_2_O_2_, 24 h) induced a significant increase of mRNA expression of 3β-HSD, 17β-HSD, 5α-R2, and aromatase (Cyp-19), compared with control cells (respectively: 61.6 ± 6.1-, *p* < 0.01; 4.6 ± 0.2-, *p* < 0.05; 1.5 ± 0.2-fold change, *p* < 0.05; 1.6 ± 0.1-fold change, *p* < 0.05) (Figure 4A–D). Pre-treatment with tadalafil reduced the expression of 5α-HSD mRNA induced by H_2_O_2_ (0.9 ± 0.1-fold change, *p* < 0.05) and significantly increased the expression of Cyp-19 (3.5 ± 0.2-fold change, *p* < 0.05) but did not significantly affect the mRNA expression of 3β-HSD and 17β-HSD. Interestingly, the presence of tadalafil per se induced a significant increase of the transcripts of all steroidogenic enzymes compared with control cells (respectively, 28.4 ± 0.6-, *p* < 0.01; 6.5 ± 0.8-, *p* < 0.05; 2.8 ± 0.3-fold change, *p* < 0.05, 1.2 ± 0.1-fold change, *p* < 0.05); moreover, it showed a stronger effect of H_2_O_2_ on 17β-HSD and 5α-R2 mRNAs (*p* < 0.05).

## 3. Discussion

Our findings offer compelling lines of evidence indicating that moderate levels of H_2_O_2_ can induce modulation of steroidogenesis in muscle cells, as indicated by the increased expression of 3β-HSD, 17β-HSD 5α-R2, and aromatase, as well as by the release of DHT. We believe that this activation can contribute to the adaptation process to an oxidizing environment, such as that caused by strenuous exercise, and considering the data from the literature, probably ameliorates cell survival. Tadalafil per se induced a significant increase in the transcripts of all steroidogenic enzymes; however, this effect was followed by a small release of DHT, while the combined treatments blunted the DHT release induced by H_2_O_2_.

To verify that we were performing experiments in non-cytotoxic conditions, we evaluated the expression of CRYAB, a protein sensitive to the alteration of redox homeostasis, along with apoptotic markers after treatments.

CRYAB expression appears to be dependent on the levels of ROS (cytotoxic or non-cytotoxic) [23,24,25,26,27]. In our experiments, we did not observe upregulation of CRYAB protein expression, and no featured changes related to cell death were observed at the morphological and molecular levels. On the contrary, we observed a significant increase in the phosphorylated form that is known to participate early in the oxidative stress adaptive response of skeletal and cardiac muscles [25,26]. Depending on the type and/or duration of various stimuli, a part of the CRYAB pool becomes phosphorylated and, correlatively, shows an enhanced affinity for the various elements of the cell, providing beneficial outcomes [28]. These data confirm that our experimental conditions induced eustress in C2C12 myotubes.

### 3.1. H_2_O_2_ Induces the Release of DHT but Not Testosterone

In our experimental model, H_2_O_2_ induced the release of DHT but not testosterone. Dihydrotestosterone is a metabolite of testosterone that is produced in many tissues, following the rapid and irreversible reduction in testosterone by 5α-reductase. It has been reported that DHT, but not T, modulates force production in isolated, intact, mouse skeletal muscle fibers, and stimulates amino acid uptake [29,30]. Moreover, several works have demonstrated the protective effect of DHT against oxidative stress, in both in vitro and in vivo experimental models.

In a model of mouse embryonic stem cells, DHT pre-treatment prevented H_2_O_2_-induced cell injury through inhibition of ROS and ROS-induced activation of different signaling pathways, such as p38-mitogen-activated protein kinase (p38MAPK) and stress-activated protein kinase (SAPK)/JNK and NF-κB) [31]. DHT treatment reduced amyloid-beta peptide 1–42 (Aβ1–42)-induced oxidative stress, and the internalization of Aβ1–42 by Z310 cell line epithelial cells [32]. It was demonstrated that DHT enhanced resistance to oxidative, stress-induced apoptosis on endometrial stromal cells by enhancing forkhead box protein O1 (FOXO1) expression, in parallel with increased manganese-dependent superoxide dismutase (SOD2) [33]. DHT was also found protective in pancreatic islet INS-1 β-cells against H_2_O_2_-induced oxidative stress [34]. Moreover, the inhibition of DHT production through treatment with finasteride, an inhibitor of 5α-reductase, induced oxidative stress in an experimental rat model, indicating an autocrine protective role of dihydrotestosterone [35].

In our results, we found that stimulation of C2C12 cells with H_2_O_2_ induced the release of DHT but did not influence T release. To explain the significance of the different effects of H_2_O_2_ on the release of the two hormones, we speculate that H_2_O_2_ itself, at a non-cytotoxic level, produces a defensive microenvironment, thus protecting the secreting cell itself, as well as nearby cells. Instead, it is possible that increased T levels might lead to an alteration in the balance between ROS and antioxidant defenses and, therefore, to an enhanced risk of oxidative stress. In fact, several studies have demonstrated that T has mainly pro-oxidant properties in different tissues, including muscles, especially if it is present at high levels [36,37,38,39,40].

In this context, we can suppose that DHT could be part of a defensive communication system between cells that allows them to adapt to changing redox environmental conditions [41].

### 3.2. H_2_O_2_ Modulates the Expression of Steroidogenic Enzymes

In the past, it was believed that only gonads were responsible for androgen production. At present, it has been demonstrated that other tissues are able to produce androgens. Recently, muscle cells have been shown to possess steroidogenesis enzymes, whose activity seems to be modulated by several physiological stimuli [42]. In particular, 17β-hydroxysteroid (17β-HSD), 3β-hydroxysteroid (3β-HSD), 5α-reductase, and P450 aromatase are particularly expressed in the ovary, in testis, and in the brain, but also in muscles, such as the gastrocnemius, which presents the highest expression of 3 β-HSD, and the soleus, with the highest expression of P450 aromatase [42]. Although studies conducted on C2C12 revealed the enzymatic activity of 3β-HSD, the presence of the protein was still not detected in their results [43].

Here, we found that H_2_O_2_ exposure induced the transcription of 3β-HSD, 17β-HSD, and 5α-R2 mRNAs.

Several studies have also reported that physical activity plays a role in the steroidogenesis of muscle cells. Mechanical stimulation promoted the expression of the steroidogenic enzymes 3β-HSD, 17β-HSD, and aromatase in C2C12 cells, confirming the importance of this biochemical pathway in contracting muscles [22].

Further, it has been shown that endurance exercise training enhances muscular DHT concentration through steroidogenic enzyme modulation in human and rat skeletal muscles, suggesting that local bioactive androgen production may participate in exercise training-induced skeletal muscular adaptation [21,44,45].

The influence of exercise on steroidogenesis has been proven to be very promising; for example, it was reported that age-related declines in sex steroidogenic enzymes and muscle sex steroid hormone levels were restored via a progressive resistance training program through 3β-HSD, 17β-HSD, and 5α-reductase expression inductions [3]. Indeed, a rapid increase in serum DHT concentration has been observed immediately after repeated, high-intensity sprint exercises in healthy subjects [46].

Dehydroepiandrosterone (DHEA) is a precursor of sex steroid hormones, and in vivo, its circulating levels provide substrates required for conversion into androgens and estrogens in peripheral tissues [47]. Our experiments confirmed that the presence of exogenous DHEA, such as that contained in the bovine serum normally used for cell culture, is essential to stimulate steroidogenesis in any experimental condition (1500 times *p* < 0.001, Table 1). In a serum-deprived condition, no DHT release was observed.

### 3.3. TAD Influences Steroidogenesis in C2C12 Myotubes

Various non-prohibited substances and nutritional supplements are often used by competitive and non-competitive athletes in order to prevent or reduce metabolic imbalance, thus improving physical performance. Some of them have been shown to influence the endogenous steroids milieu and energy metabolism, both at rest and during exercise. Among these compounds, different effects of phosphodiesterase’s type 5 inhibitor (PDE5i) tadalafil on exercise adaptation, performance, and modulation of endocrine, enzymatic, and metabolic pathways were observed [20,48,49,50,51,52,53,54]. Tadalafil amplifies the nitric oxide (NO) biological activity, thus modulating the endogenous steroids release and energy substrate metabolism, at rest and after physical stress [51,55,56,57,58].

Recently, our laboratory, along with others, has demonstrated that PDE5i, including tadalafil, may modulate muscle metabolism and response to ROS under physiological and pathological conditions, in either in vitro or in vivo studies [27,59,60,61,62,63,64,65,66].

Here, we observed a different effect of tadalafil when administered alone, or in combined treatment with H_2_O_2._ In particular, tadalafil treatment did not influence T release, while it induced a slight increase in DHT levels, along with the increase in 17β-HSD and 5α-reductase expressions.

While tadalafil, in the combined treatment, significantly blunted the DHT release, H_2_O_2_ induced and reduced the expression of 5α-R2. According to this observation, recently, we have demonstrated that tadalafil administration in vivo significantly blunted serum DHT increase after maximal aerobic exercise, compared to placebo in young men [21]. The blunting effect of tadalafil on DHT supports a possible role of peripheral nitric oxide/GMPc-related pathways in influencing physical stress-related steroidogenesis and DHT metabolism, making cells less dependent on the protective action of DHT.

Interestingly, a marked increase in aromatase expression was observed, possibly increasing the conversion of T to estradiol, a hormone also involved in response to oxidative stress in muscle cells. Therefore, it is possible that an additional mechanism exists in muscle cells that reduces the expression of the steroidogenic enzyme responsible for the conversion of T to DHT [67] and increases the conversion to estradiol in mild stress conditions. In fact, there was a demonstrated protective role of estradiol in muscle cells [68,69,70,71,72].

## 4. Conclusions

Through this study, we provided a new piece of evidence to the picture of the adaptive skeletal muscle cell response to a pro-oxidant environment.

Given that C2C12 myotubes possess most of the morpho-functional features of contractile muscle cells, and considering redox imbalance, a condition similar to what occurs during muscle contraction [73], we can speculate that our observations in vitro about the intramuscular steroidogenesis might be extended in vivo, in order to better understand muscle function and adaptation related to exercise.

This area of research is clearly in its early stages and warrants extended additional investigations in humans.

## 5. Materials and Methods

All chemical reagents, unless otherwise specified, were purchased from Sigma-Aldrich Chemical (St. Louis, MO, USA).

### 5.1. Cell Culture

C2C12 myoblasts (2 × 10^3^ cm^2^, passage number 6; ATCC, Manassas, VA, USA) were cultured in 25 cm^2^ culture flasks with Dulbecco’s modified Eagle’s medium (DMEM; HyClone, Oud-Beijerland, Holland) supplemented with Glutamax-I (4 mM L-alanyl-L-glutamine), 4.5 g/L glucose (Invitrogen, Carlsbad, CA, USA), and 10% heat-inactivated fetal bovine serum (FBS; Hy-Clone, Oud-Beijerland, Holland). The cells were incubated at 37 °C with 5% CO_2_ in a humidified atmosphere. Cells were split 1:6 twice weekly and fed 24 h before each experiment. Differentiation into myotubes was achieved via culturing pre-confluent cells (85% confluency) in a medium containing 2% FBS, and they were monitored via microscopy, and for myogenin and MHC expression, via Western blot analysis [74,75].

### 5.2. Cell Treatments

C2C12 myotubes were treated with H_2_O_2_ (500 µM) alone or after pre-treatment with tadalafil (1 µM, 30 min) and in combined treatments for the following 1, 6, and 24 h.

MTT assay was performed, and no statistically significant differences were found after H_2_O_2_ treatment (data not shown).

The same experiments were also performed using a medium without phenol red and FBS using DHEA (500 nM) as positive control. Each experiment was performed in triplicate.

### 5.3. Morphological Imaging

Morphological changes and cell apoptosis were assessed by analyzing photomicrographs obtained under an inverted phase-contrast microscope (Nikon Eclipse TS100. Nikon Europe BV, Amsterdam, The Netherlands) with a digital camera (Canon Europe, Amstelveen, The Netherlands). Cellular analysis was performed by evaluating at least five different fields for each well.

### 5.4. Protein Expression Analysis

C2C12 myotubes with or without tadalafil (1 µM), and then stimulated for 1 h or 24 h with 500 µM H_2_O_2,_ were lysed in RIPA buffer (150 mM NaCl, 50 mM Tris-HCl pH 8, 1 mM EDTA, 1% NP40, 0.25% sodium deoxycholate, 0.1% SDS, water to volume), supplemented with protease and phosphatase inhibitor cocktails (Sigma–Aldrich, Darmstadt, Germany). As previously described [76,77,78], for the immunoblot analysis, an equal amount of proteins (20–30 µg) was resolved in SDS-polyacrylamide (BIO-RAD) gels (10–12%), and transferred onto nitrocellulose membranes (Amersham, Little Chalfont, UK). Thereafter, membranes were incubated with primary antibodies appropriately diluted in Tween Tris-buffered saline (TTBS). Proteins were revealed by the enhanced chemiluminescence system (Amersham Biosciences, GE Healthcare Europe GmbH, Glattbrugg, Switzerland). Image acquisition was performed with Image Quant Las 4000 software (GE Healthcare, Chicago, IL, USA), and densitometric analysis, with Quantity One^®^ software (Bio-Rad Laboratories, Inc., Hercules, CA, USA). The following antibodies were utilized: *p*-CRYAB, CRYAB, Bcl-2, Bax, caspase-3, and β-actin from Santa Cruz (Santa Cruz Biotechnology, Santa Cruz, CA, USA).

### 5.5. Testosterone and Dihydrotestosterone Levels

Following each experimental point, the culture medium was collected and stored at −20 °C until it was assayed for DHT and testosterone (T). DHT concentration was measured via enzyme-linked immunosorbent assay, using commercial kits (DRG International Inc., Marburg, Germany).

T concentration was measured via radioimmunoassay, using commercial kits (Immunotech, Radiova, Prague, Czech Republic, and Orion Diagnostica Oy, Espoo, Finland). All samples were analyzed in duplicate within the same assay. The sensitivity of the method was 7.23 pg/mL for DHT, and 0.1 nmol/L for T. The coefficients of variation for intra-assays and inter-assays were 6.25% and 7.47% for DHT, and 7.5% and 7.0% for T, respectively.

### 5.6. RNA Extraction, Reverse Transcription, and Real-Time Quantitative PCR

Total RNA was obtained from ≈3.5 × 10^4^ cells using TRIZOL, according to the manufacturer’s instructions and as previously described [65,79]. Treatment with DNase enzyme was performed to remove genomic DNA contamination. cDNA was obtained via reverse transcription of 500 ng of total RNA. RT-qPCRs were performed, as previously described [24,80]. Fluorescence intensities were analyzed using the manufacturer’s software (7500 Software v2.05, Applied Biosystem, Waltham, MA, USA), and relative amounts were evaluated using the 2−∆Ct method and normalized for β-actin. Data are expressed as a fold increase. Sequences of primers are shown in Table 2.

### 5.7. Statistical Analysis

All data are expressed as means ± SE of three independent experiments, each performed in triplicate. A one-way ANOVA and Bonferroni post hoc analysis were used to determine significant variations among groups for each parameter evaluated; *p* < 0.05 was accepted as significant. The SPSS statistical package (Version 17.0 for Windows; SPSS Inc., Chicago, IL, USA) was used for statistical analysis.

## Figures and Tables

**Figure 1 ijms-23-06566-f001:**
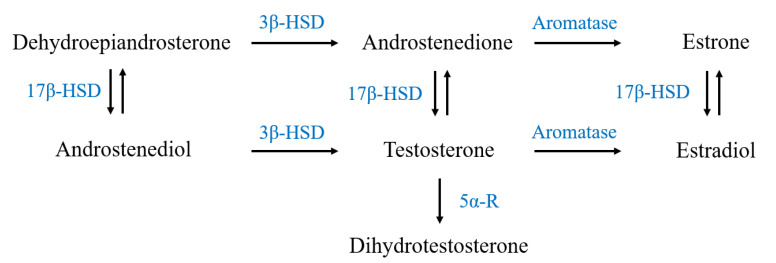
Schematic representation of the steroidogenesis pathway: 3β-HSD: 3β-hydroxysteroid dehydrogenase; 17β-HSD: 17β-hydroxysteroid dehydrogenase; 5α-R: 5α-reductase.

**Figure 2 ijms-23-06566-f002:**
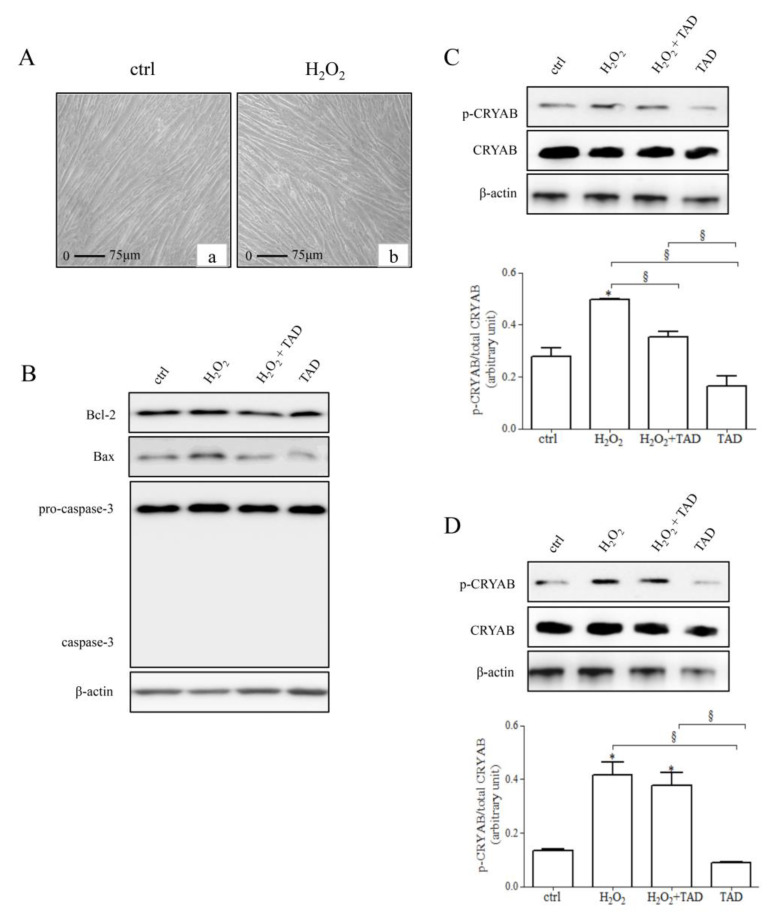
(**A**) Morphological analysis of cell death in control C2C12 myotubes (left panel, a), or exposed to 500 µM H_2_O_2_ for 24 h (right panel, b). Scale bar, 75 μm. (**B**) Representative immunoblot images of specific apoptotic markers, such as Bcl-2, Bax, and caspase-3, were evaluated in C2C12 myotubes exposed to hydrogen peroxide (500 µM for 24 h) in presence or not of tadalafil (1 µM, 30 min pre-treatment, and then combined treatments for the following 24 h). (**C**,**D**) Representative immunoblot images of total and phosphorylated form of CRYAB, evaluated in C2C12 myotubes exposed to hydrogen peroxide (500 µM) for 1 h (**C**) or 24 h (**D**), pre-treated, or not, with tadalafil. Bar diagrams represent the densitometric intensities of *p*-CRYAB, normalized with total CRYAB. β-actin was used as loading control. * *p* < 0.05 vs. ctrl; ^§^ *p* < 0.05 vs. H_2_O_2_.

**Figure 3 ijms-23-06566-f003:**
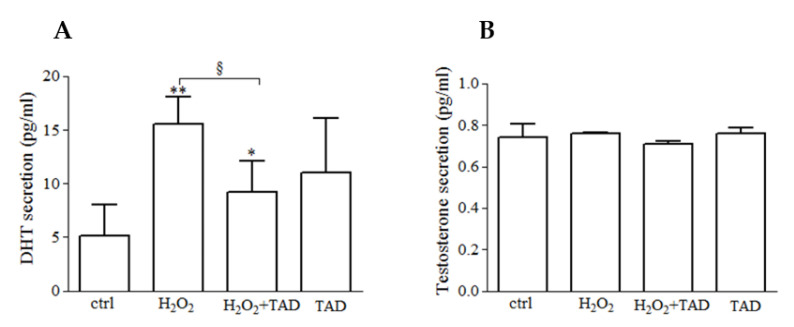
C2C12 myotubes exposed to hydrogen peroxide (500 µM, 24 h) in presence or absence of tadalafil (1 µM, 30 min pre-treatment, and then combined treatments for the following 24 h) were analyzed for DHT (**A**) and T (**B**) release. Data are presented as the percentage of release vs. ctrl ± SE. * *p* < 0.05 and ** *p* < 0.01 vs. ctrl; ^§^
*p* < 0.05 vs. H_2_O_2_.

**Figure 4 ijms-23-06566-f004:**
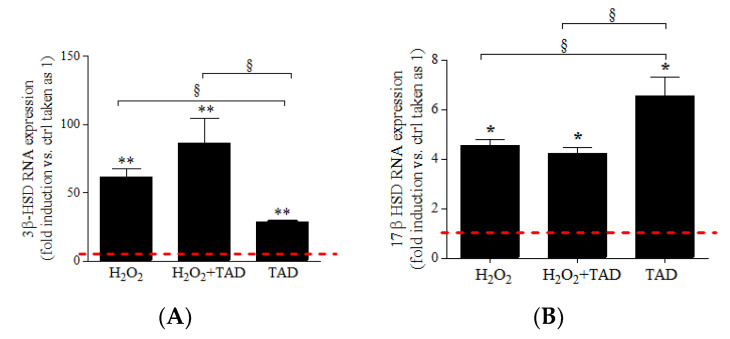
(**A**) 3β-HSD, (**B**) 17β-HSD, (**C**) 5α-R2, and (**D**) aromatase (Cyp-19) mRNA of C2C12 myotubes exposed to H_2_O_2_ (500 µM, 24 h for Cyp-19) in presence or absence of tadalafil (1 µM, 30 min pre-treatment and then combined treatments for the following 24 h). Data are shown as the fold increase vs. ctrl taken as 1 (red line) ± SE (n = 3). * *p* < 0.05 and ** *p* < 0.01 vs. ctrl; ^§^ *p* < 0.05 vs. H_2_O_2_.

**Table 1 ijms-23-06566-t001:** DHT evaluation in serum-free condition after H_2_O_2_ and TAD administration.

	DHT (pg/mL)
ctrl	n.d.
H_2_O_2_	n.d.
H_2_O_2_ + TAD	n.d.
TAD	n.d.
DHEA	1547.5 ± 395.7 *

* *p* < 0.001 vs. ctrl; n.d. not detectable.

**Table 2 ijms-23-06566-t002:** Sequences of primers for RT-PCR analysis.

Gene Name	Forward (5′-3′)	Reverse (5′-3′)
** *5α-R2* **	TGGAGGGCATGGTGCTAAAG	TCTCTCACTTAGCACGGGGA
** *17β-HSD* **	TTTGCGCTCGAAGGTTTGTG	GCAGTCAAGAAGAGCTCCGT
** *3β-HSD* **	ACCTTGTGGCTGACCATCTC	TGCTCTTCCTCGTTGCCATT
** *CYP-19* **	AACCCCATGCAGTATAATGTCAC	AGGACCTGGTATTGAAGACGAG
** *β-actin* **	CTGAACCCCAAGGCCAAC	AGCCTGGATAGCAACGTACA

## Data Availability

The data presented in this study are available on request from the corresponding author.

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
