# Peer review of "Hydrogen Peroxide Stimulates Dihydrotestosterone Release in C2C12 Myotubes: A New Perspective for Exercise-Related Muscle Steroidogenesis?"

_ijms, 2022, doi:10.3390/ijms23126566_

Round 1

Reviewer 1 Report

This study reports very interesting and unique data on the steroidogenesis process in muscle cells. The study is well-designed and described.

Before further processing, only minor modifications are needed.

1.     “in reproductive Leydig cells” rather testicular steroidogenic Leydig cells”

2.       provide bars for microphotographs

3.       give passage number of used cell line

4.       H2O2 is used to induce cell senescence in vitro- is it possible that you also activated this process? Give the rationale for used treatment dose

Author Response

Replies to Reviewer 1

This study reports very interesting and unique data on the steroidogenesis process in muscle cells. The study is well-designed and described.

Before further processing, only minor modifications are needed.

We thank the Reviewer for his positive appreciations and comments that help us improve the quality of our manuscript.

All changes made in the revised version of the manuscript are highlighted in red

As request, the manuscript was checked by a native English speaker.

Replies to comments are listed below.

Q1 … “in reproductive Leydig cells” rather testicular steroidogenic Leydig cells”

We thank the Reviewer for the comment. We changed the text accordingly.

Q2 … provide bars for microphotographs

We thank the Reviewer for the note. We changed the text accordingly.

Q3 … give passage number of used cell line

We thank the Reviewer for the note. We changed the text accordingly.

Q4 … H2O2 is used to induce cell senescence in vitro- is it possible that you also activated this process? Give the rationale for used treatment dose

We thank the Reviewer for the comment.

In this work we treated myotubes with 500 mM H2O2. In these experimental conditions we did not observe cell death, nor alteration in the number/size of myotubes.

The analysis of the levels of CRYAB, a protein that is activated after redox imbalance through phosphorylation, and the analysis of the apoptotic markers showed that hydrogen peroxide induced a redox perturbation (activation of CRYAB phosphorylation) without involvement of Bax, Bcl-2 and Caspase 3 cleavage, perfect conditions for the purpose of our work, that was to verify the modulation of steroidogenesis in conditions of mild cellular stress.

Reviewer 2 Report

The work concerns the influence of free radicals on the dihydrotestosterone of C2C12 myoblast cells.

The aim of this study was to investigate whether ROS could influence steroidogenesis in myoblast cells by evaluating the release of testosterone and dihydrotestosteron as well as the evaluation of expression of key steroidogenic enzymes 5α-reductase, 3β-hydroksysteroid dehydrogenase, 17β- hydroksysteroid dehydrogenase, aromatase.

The obtained results bring new information to the existing knowledge.

For first-time readers of these topics, Figure 1 is beneficial.

Wouldn't it be better to write in the title instead of C2C12 muscle cells - C2C12 myoblasts.

In the materials and methods it was written that 20-30 mg of protein were taken for the experiments, if there are such differences by one third of the amount, then whether or not it introduces errors. In figure 2 C, D, looking at the amount of actin, there appears to be less of the administered protein in the TAD group due to whether the results presented in the figure are converted into the amount of actin.

Table 1: Isn't it better to use a larger unit of ng / ml.

How can you explain the lack of increase in testosterone concentration after the administration of H2O2, if the expression of enzymes influencing its synthesis increases.

Author Response

Replies to Reviewer 2

The work concerns the influence of free radicals on the dihydrotestosterone of C2C12 myoblast cells.

The aim of this study was to investigate whether ROS could influence steroidogenesis in myoblast cells by evaluating the release of testosterone and dihydrotestosteron as well as the evaluation of expression of key steroidogenic enzymes 5α-reductase, 3β-hydroksysteroid dehydrogenase, 17β- hydroksysteroid dehydrogenase, aromatase.

The obtained results bring new information to the existing knowledge.

For first-time readers of these topics, Figure 1 is beneficial.

We thank the Reviewer for his positive appreciations and comments that help us improve the quality of our manuscript.

All changes made in the revised version of the manuscript are highlighted in red

As request, the manuscript was checked by a native English speaker.

Replies to comments are listed below.

Q1 … Wouldn't it be better to write in the title instead of C2C12 muscle cells - C2C12 myoblasts.

We thank the Reviewer for the comment. We changed the text accordingly instead “C2C12 muscle cells” we specify “C2C12 myotubes”.

Q2 … In the materials and methods it was written that 20-30 mg of protein were taken for the experiments, if there are such differences by one third of the amount, then whether or not it introduces errors. In figure 2 C, D, looking at the amount of actin, there appears to be less of the administered protein in the TAD group due to whether the results presented in the figure are converted into the amount of actin.

We thank the Reviewer for this comment. In the manuscript we have specified that we used 20-30 mg of protein because protein loading depends on the expected expression of the protein target. In fact, is a good practice to load a specific amount of proteins in each western blot dependently from their expression. In our Figure 2, we showed a representative image for all targets, including the selected housekeeping (beta-actin). All targets have been normalized versus beta-actin.

In different experiments, for each lane in the electrophoretic run the same amount of sample was loaded and the target protein normalized with beta actin, the housekeeping used as internal control.

The images are therefore purely representative, while the histogram graph is the result of all the experiments carried out.

Q3 … Table 1: Isn't it better to use a larger unit of ng / ml.

We thank the Reviewer for the comment. usually, the unit of measurement for DHT values is in the order of pg/ml. By maintaining the same unit of concentration with respect to Figure 3, it is also possible to appreciate the magnitude of increase induced by the treatment with DHEA.

Q4 … How can you explain the lack of increase in testosterone concentration after the administration of H2O2, if the expression of enzymes influencing its synthesis increases.

We thank the Reviewer for the comment. We have included the explanation of this difference in the discussion section at lines 232-243 “In our results, we found that stimulation of C2C12 cells…..”.

Furthermore, it must be considered that the level of expression does not always correlate with the level of enzymatic activity so we can only speculate that probably in myotubes there is a compensatory system that favors the production of DHT rather than T.